# Shift Pose: A Lightweight Transformer-like Neural Network for Human Pose Estimation

**DOI:** 10.3390/s22197264

**Published:** 2022-09-25

**Authors:** Haijian Chen, Xinyun Jiang, Yonghui Dai

**Affiliations:** 1The College of Information, Mechanical and Electrical Engineering, Shanghai Normal University, Shanghai 201418, China; 2Management School, Shanghai University of International Business and Economics, Shanghai 201620, China

**Keywords:** human-computer interaction, real-time human pose estimation, shift operator, transformer, residual log-likelihood estimation, regression-based approach

## Abstract

High-performing, real-time pose detection and tracking in real-time will enable computers to develop a finer-grained and more natural understanding of human behavior. However, the implementation of real-time human pose estimation remains a challenge. On the one hand, the performance of semantic keypoint tracking in live video footage requires high computational resources and large parameters, which limiting the accuracy of pose estimation. On the other hand, some transformer-based models were proposed recently with outstanding performance and much fewer parameters and FLOPs. However, the self-attention module in the transformer is not computationally friendly, which makes it difficult to apply these excellent models to real-time jobs. To overcome the above problems, we propose a transformer-like model, named ShiftPose, which is regression-based approach. The ShiftPose does not contain any self-attention module. Instead, we replace the self-attention module with a non-parameter operation called the shift operator. Meanwhile, we adapt the bridge–branch connection, instead of a fully-branched connection, such as HRNet, as our multi-resolution integration scheme. Specifically, the bottom half of our model adds the previous output, as well as the output from the top half of our model, corresponding to its resolution. Finally, the simple, yet promising, disentangled representation (SimDR) was used in our study to make the training process more stable. The experimental results on the MPII datasets were 86.4 PCKH, 29.1PCKH@0.1. On the COCO dataset, the results were 72.2 mAP and 91.5 AP50, 255 fps on GPU, with 10.2M parameters, and 1.6 GFLOPs. In addition, we tested our model for single-stage 3D human pose estimation and draw several useful and exploratory conclusions. The above results show good performance, and this paper provides a new method for high-performance, real-time attitude detection and tracking.

## 1. Introduction

Human pose estimation (HPE) is a classical computer vision problem, with longstanding studies, that aims to infer articulated structures of human parts from a single image. Existing approaches can be divided into two categories: heatmap-based [1,2,3,4,5,6,7,8,9,10,11] and regression-based [12,13,14,15,16,17,18]. The former usually adapts convolution neural networks as deep features extractor and then generates the heatmap of joints with deconvolution layers. The latter directly regresses the numerical coordinates of joints with fully connected neural networks or others.

Recently, more and more human-centered applications with real-time requirements, such as self-driving and last-mile delivery robots, emerge in large numbers. However, existing models are either heatmap-based with high accuracy and low speed or regression-based with high speed and low accuracy. For example, HRNet-W48 [19] archives 75.6 mAP on the COCO [20] dataset with more than 63M parameters, 15.77 FLOPs, less than 22 fps on GPU, about 1.46 fps on the CPU. DeepPose (ResNet-50) [21] can run at 135 fps, but only archives 52.6 mAP on the COCO dataset.

The regression-based approach is simpler and more computationally friendly than the heatmap-based approach. However, numerical regression tends to lack spatial generalization and robustness; more importantly, the regression-based approach does not performer better than the traditional heatmap-based approach. These problems attract lots of researchers to propose effective and efficient solutions [14,17,18].

Recently, a number of outstanding transformer-based [22] works have emerged and received excellent performance with less parameters [23,24,25]. However, the self-attention module has high computational complexity, which goes against our goal: real-time. Fortunately, recent works show that the attention-based module in transformers can be replaced by some simple modules, and even nonparameterized operations still perform quite well [26,27,28,29,30].

In this paper, we explore the efficient human pose estimation model for real-time tasks and propose our lightweight model: ShiftPose. A previous study [19] inspired us to introduce high resolution and multi-branches into our model, which achieves good performance on human pose estimation. Meanwhile, researchers also found that multi-branch structure is redundant in lightweight models [31]. Based on above findings, we designed a simple and efficient model that adapts ShiftViT [29] as the backbone and introduces bridge–branch structure. On the MPII dataset, we had 86.4 PCKH, 29.1 PCKH@0.1. On COCO dataset, we had 72.1 mAP, 255 fps on the GPU with 10.2M parameters, and 1.6 GFLOPs.

The contributions of our model can be summarized as follows:We propose a simple and efficient transformer-like model, without the self-attention module for HPE. The proposed model has few parameters, lossless accuracy, and runs much faster than existing transformer-based models [32,33].An improve residual log-likelihood estimation loss is proposed, and we apply it to 3D human pose estimation.Our model is competitive with the heatmap-based model and even better than heatmap-based model for indicating AP50.We first find that, with a restricted number of parameters, the lightweight model tends to learn the x- and y-coordinates as priority in 3D human pose estimation, which points toward the direction to improve the performance of future lightweight models in 3D human pose estimation.

## 2. Relative Work

### 2.1. Regression-Based HPE

Before deep learning had a huge impact on vision-based human pose estimation, traditional 2D HPE algorithms adopted handcraft feature extraction and sophisticated body models to obtain local representations and global pose structures [34,35,36].

There are only a few regression-based works, in the context of human pose estimation. DeepPose firstly uses AlexNet-like convolution neural network to learn joint coordinates from a single image [21]. Luvizon proposes a soft-argmax function to convert a heatmap to a numerical joint position, which makes the model differentiable and more robust [12]. Another important work is DSNT, which makes the model differentiable and performs well on low resolution input [13]. In order to make regression learning easier, some researchers try to improve the training process, such as an iterative error feedback network [14], and other researchers adapt a multi-tasking framework as their training paradigm [15,16]. In 3D HPE, researchers tend to first use a heatmap-based method to learn 2D joints coordinates, and then learn depth information separately [37,38,39,40,41]. Recently, residual log-likelihood estimation (RLE), which makes the regression-based approach perform well, or even better, than heatmap-based approach [18].

### 2.2. Lightweight Model

Mobilenet [42] proposes the depthwise separable convolutions, and Mobilenetv2 [43] introduces the inverted residual with linear bottleneck. Both MobileNet and MobileNetv2 improve the computational efficiency of convolution operations. ShuffleNet [44] reduces computation with pointwise group convolution and channel shuffle operation. Repvgg [45] converts the multi-branch structure to a single-branch structure with the reparametrization trick, thus improving the inference efficiency of the model.

Lite-hrnet [46] applies the improved shuffle blocks to HRNet, but it only gets 12 fps on GPU. Lite pose [31] finds that HRNet’s high-resolution branches are redundant for models at the low-computation region via gradual shrinking experiments. Additionally, the bridge–branch structure is inspired by this finding.

### 2.3. Transformer

With the success of vision (ViT) [24], Swin [23], and data efficient image (DeiT) [25] transformers in computer vision, more and more scholars adapt the vision transformer as their backbone and achieve outstanding performance in their tasks.

Token Pose [33] firstly applied a pure transformer to human pose estimation. HRFormer [32] replaces the block in HRNet with transformer-like block, which gains higher accuracy, with less parameters, than DeiT. Without exceptions, all the models mentioned above are heatmap-based and computationally unfriendly.

The recent research shows that the attention-based module in transformers can be replaced by some simple modules, and even nonparameterized operations still perform quite well. What’s more, the self-attention module in transformers costs large computation and video memory. gMLP [28] replaces the self-attention module in the transformer with spatial MLPs and still works very well. MetaFormer [30] deliberately replaces the attention module in the transformers with a pooling operator to conduct only basic token mixing. Surprisingly, it still achieves competitive performance on multiple computer vision tasks. ShiftViT [29] is a Swin transformer-like model that simply removes the self-attention module and uses the shift operator instead, which also gets the competitive results. We suggest future works should pay more attention to the other modules in transformers, such as LayerNorm, feed forward networks, and so on.

To the best of our knowledge, this paper is the first to introduce ShiftViT into a regression-based model; we apply it to 2D HPE and 3D HPE and gain high accuracy and efficient results.

### 2.4. Real-Time Human Pose Estimation

Lite pose [31] explores efficient architecture design for real-time, multi-person pose estimation on resource constrained edge devices and reveals that HRNet’s high-resolution branches are redundant for models at the low-computation region via the gradual shrinking experiments. OpenPose [47] proposes the part affinity fields (PAF) used to learn multi-person coordinates via the bottom-up method. Recently, the lightweight bottom-up model named Lite-Pose [31], for the first time, discovered that HRNet’s high-resolution branches are redundant for models at the low-computation region, which is also one of our motivations.

## 3. Method

### 3.1. Overall Architecture

An overall architecture of our model is presented in Figure 1. It first splits an input image into 4×4 patches by patch partition and linear embedding module, like ViT. Then, the backbone followed by it can be divide into 5 shift stages. Each shift stage contains several shift blocks and an after-shift stage; the patch merging module will make the spatial size of the output half down-sampled, while the channel size is twice the input, and the patch making module will make the spatial size of output double up-sampled, while the channel size is half of the input.

After the spatial gate module, if the input image’s shape is denoted as $\left(C_{in}\;,\;H,\;W\right)$, the output’s shape should be $\left(n_{joints},\frac{H}{8},\frac{W}{8}\right)$. The regression head contains two simple linear layers, with $channel_{input}=\frac{H}{8}\times \frac{W}{8}$ and $channel_{outputX}=H$ for X’s coordinate and $channel_{outputY}=W$ for Y’s coordinate.

### 3.2. Bridge–Branch Connection

To fully utilize the benefits of multi-resolution with less computation, we add the feature from early stage and feature from late stage with a bridge structure. This simple skip-connection performs quiet well in our model.

Specifically, as shown in Figure 2, the output of the first patch merging module will be sent to the spatial gate module, and the output of the second patch merging module will be sent to shift stage 3. This residual-like structure can make full use of the feature from each stage of the model, which makes up for the disadvantages of a few parameters.

### 3.3. Shift Operator

As shown in Figure 3, our shift stage is similar to ShiftViT [29]; however, in the HPE task, the channel of the output is much smaller than ShiftViT. In order to make full use of the output feature, we add a SE layer [48] at the end of the shift block. The shift operation is cheap and effective, which reduces quiet a lot FLOPs in training and testing.

The shift operator can be formulated as follows:(1)$$\hat{z}\left[0:H,1:W,0:\gamma C\right]\leftarrow z\left[0:H,0:W-1,0:\gamma C\right],$$
(2)$$\hat{z}\left[0:H,0:W-1,\gamma C:2\gamma C\right]\leftarrow z\left[0:H,1:W,\gamma C:2\gamma C\right],$$
(3)$$\hat{z}\left[0:H-1,0:W,2\gamma C:3\gamma C\right]\leftarrow z\left[1:H,0:W,2\gamma C:3\gamma C\right],$$
(4)$$\hat{z}\left[0:H,0:W,3\gamma C:4\gamma C\right]\leftarrow z\left[0:H-1,0:W,3\gamma C:4\gamma C\right],$$
(5)$$\hat{z}\left[0:H,1:W,4\gamma C:C\right]\leftarrow z\left[0:H,0:W,4\gamma C:C\right]$$
where the input $\hat{z}\in R^{H\times W\times C}$. In our experiments, γ = 1/12, which is same as [28].

### 3.4. Patch Merging and Patch Making

The patch merging module merges neighboring patches through the convolution with a kernel size of 2 × 2. After patch merging, the spatial size of the output is half down-sampled, while channel size is twice the input, i.e., from C to 2C.

On the contrary, the patch merging module creates patches through the deconvolution with a kernel size of 2 × 2. After patch making, the spatial size of the output is half up-sampled, while channel size is half of the input, i.e., from 2C to C.

### 3.5. Spatial Gate

At the end of our model, the channel of output is a bit large. It is a waste of time to use the MLP to reduce the channel, so we introduce the *Spatial Gate Unit* from [28] to reduce the channel. The detailed structure is shown in Figure 4.

First, we reshape the output $\left(C,H,W\right)$ to $\left(C,H\times W\right)$; after the proj_in and split operation, it becomes $X_{1}\left(\frac{C}{2},H\times W\right)$ and $X_{2}\left(\frac{C}{2},H\times W\right)$. The *spatial proj* module can be formulated as follows:(6)$$f_{W,b}\left(X\right)=WX+b,$$
where $W\in R^{H\times W}$ is the spatial project matrix.

The final output can be described as:(7)$$Z=X_{1}⨀f_{W,b}(X_{2})$$

### 3.6. SimDR

Directly regressing the numerical coordinates lacks spatial generalization and robustness, resulting in inferior predictions in most tough cases. To make it easier for our model to learn, we apply the SimDR to our model training process. The simple, yet promising, disentangled representation for keypoint coordinates (SimDR) alleviates the problem of the regression-based approach from the classification point of view [27].

The coordinate will be expressed as
(8)$$X=\left[x_{0},x_{1},\cdots ,x_{W\cdot k-1}\right]\in R^{W\cdot k},x_{i}=\frac{1}{\sqrt{2\pi \sigma }}exp\left(-\frac{{\left(i-x^{\prime }\right)}^{2}}{2\sigma^{2}}\right),$$
(9)$$Y=\left[y_{0},y_{1},\cdots ,y_{H\cdot k-1}\right]\in R^{H\cdot k},x_{i}=\frac{1}{\sqrt{2\pi \sigma }}exp\left(-\frac{{\left(i-y^{\prime }\right)}^{2}}{2\sigma^{2}}\right),$$
where $x_{i}$ means the probability of appearing in position $i\in \left(0,1,\cdots ,W\cdot k-1\right),\;k$ is the scaled ratio, W is the width of the image, and the target coordinate representation is generated by Gaussian distribution. We use Kullback–Leibler divergence as loss function for model training.

The final predicted absolute joint position $\left(x_{pred},y_{pred}\right)$ is calculated by:(10)$$x_{pred}=\frac{argmax\left(X\right)}{k},y_{pred}=\frac{argmax\left(Y\right)}{k}$$

### 3.7. Residual Log-Likelihood Estimation

In 3D human pose estimation, depth estimation from one or multiple RGB images is an ill-posed problem, so it is hard to decide the length of depth representation in SimDR. Finally, we attempt to directly regress the depth of a single image with improved residual log-likelihood estimation loss [18].

As shown in Figure 5, the basic model learns the joints’ coordinate, and the flow model in the gray dotted bordered rectangle will learn the confidence of the output. In order to reduce the dependency between basic and flow models, the output of flow model will be $\log\frac{P\left(\bar{x}\right)}{s\cdot Q\left(\bar{x}\right)}$, and the constant s is to make sure this residual term is a distribution.

The original residual log-likelihood estimation is defined as follows:(11)$$Loss_{RLE}=-logQ\left({\bar{\mu }}_{g}\right)-\log G_{\phi }\left({\bar{\mu }}_{g}\right)-logs+log\hat{\sigma },$$
where $Q\left({\bar{\mu }}_{g}\right)$ is a Gaussian distribution $\left(N\left(0,1\right)\right)$, $G_{\phi }\left({\bar{\mu }}_{g}\right)$ is the distribution learned by the flow model, $s=\frac{1}{\int G_{\phi }\left({\bar{\mu }}_{g}\right)Q\left({\bar{\mu }}_{g}\right)d{\bar{\mu }}_{g}}$, which can be approximated by the Riemann sum, and $\hat{\sigma }$ is the prediction confidence. More details can be find in [18].

In our experiments, we find that, if we add a factor before $Q\left({\bar{\mu }}_{g}\right)$,
(12)$$Loss_{RLE-i}=-\gamma logQ\left({\bar{\mu }}_{g}\right)-\log G_{\phi }\left({\bar{\mu }}_{g}\right)-logs+log\hat{\sigma },$$

We set the γ=2.5.

## 4. Experiments

### 4.1. Details and Environment


**Implement details**


For the basic settings, we chose the Adam optimizer with an initial learning rate 0.001. Additionally, the learning rate was dropped to $10^{-4}$ and $10^{-5}$ at the 190th and 200th epochs, respectively. The batch size was 128, and the training epoch was 210.

On the Human3.6M dataset, we adapted the 2D and 3D mixed data training strategy for 140 epochs in total. The test procedure is the same as the previous.

The hardware for experiments includes CPU: Intel(R) Xeon(R) CPU E5-2686 v4 @ 2.30 GHz, GPU: NVIDIA RTX A4000, RAM: 240 GB. The developing environment is Ubuntu18.04, Python 3.8, CUDA 11.3, cuDNN 8, NVCC, Pytorch 1.11.0, torchvision 0.12.0, torchaudio 0.11.0.


**Notations**


Our backbone contains several stages, and each stage consists of several shift blocks, which are denoted as $M\left(B_{1},B_{2},B_{3},\cdots ,B_{n}\right)$, where M means the backbone, n means the number of stages, and $B_{i}$ means the number of shift blocks corresponding to its stage.

We provide two configurations for ShiftPose, as follows:

Shift-Pose-T(tiny) M(2,2,2,2,2) with input channel = 32;

Shift-Pose-M(mid) M(4,4,4,4,4) with input channel = 32;

Shift-Pose-L(large) M(4,4,4,4,4) with input channel = 64.

### 4.2. Dataset and Metric

**MPII Dataset [49]**: The MPII Human Pose dataset is a state of the art benchmark for the evaluation of articulated human pose estimation. The dataset includes around 25K images containing over 40K people with annotated body joints. The images were systematically collected using an established taxonomy of every day human activities. Overall, the dataset covers 410 human activities, and each image is provided with an activity label. Each image was extracted from a YouTube video and provided with preceding and following un-annotated frames. The training set contained 28,821 images, and the test set contained 11,701 images. Data augmentation on MPII dataset includes random scale [0.75,1.25], rotation degrees in [−30°,30°], and flip

**COCO Dataset [20]**: The COCO dataset contains more than 200,000 images and 250,000 person instances labeling with 17 keypoints. The COCO dataset consists of three parts: 57k images for the training set, 5k for the val set, and 20k for the test-dev set. Our method reported in this paper is trained on the train2017 set and evaluated on the val2017 and test-dev2017 sets. Data augmentation on the COCO dataset includes random scale [0.75,1.25], rotation degrees in [−30°,30°], and flip.

**Human3.6M Dataset [50]**: The Human3.6m dataset contains 3.6 million 3D human poses and corresponding images generated by 11 professional actors (6 male, 5 female) in 17 scenarios (e.g., discussion, smoking, taking photo, talking on the phone). For the Human3.6M dataset, data augmentation included random scale (±30%), rotation (±30^◦^), color (±20%), and flip. Following typical protocols [51,52], we used (S1, S5, S6, S7, and S8) for training and (S9, S11) for evaluation.

**Metric:** The percentage of porrect keypoints (PCK) was used for performance evaluation on MPII dataset. PCKh@0.5 defines the matching threshold as 50% of the head segment length, PCKh@0.1 defines the matching threshold as 10%, and the standard evaluation metric for COCO dataset is based on object keypoint similarity (OKS):$$OKS=\frac{\sum_{i}\exp\left(-\frac{d_{i}^{2}}{2s^{2}k_{i}^{2}}\right)\delta (v_{i}>0)}{\sum_{i}\delta (v_{i}>0)}$$
where $d_{i}$ is the Euclidean distance between the detected keypoint and corresponding ground truth, $v_{i}$ is the visibility flag of the ground truth, s is the object scale, and $k_{i}$ is a per-keypoint constant that controls falloff. Additionally, we used the standard average precision (AP) and recall scores: AP50 (AP at OKS = 0.50), mAP (the mean of AP scores at 10 positions, OKS = 0.50, 0.55, …, 0.90, 0.95).

For the Human3.6M dataset, the evaluation metric is mean per joint position error (MPJPE), and PA-MPJPE. PA-MPJPE is a modification of MPJPE with Procrustes analysis.

### 4.3. Results

#### 4.3.1. Result on COCO Dataset

The experiment results of our method and several of the latest transformer-based methods for human pose estimation on the COCO dataset are shown in Table 1. Some methods have different or special data preprocessing methods, although the difference between different preprocessing method is not great; for the sake of fairness, we only compare the time of inference with other methods.

As for the mean average of precision, the ShiftPose archived 72.1 AP, which was an increase of 1.7, compared to the heatmap-based method ResNet 50, 1.7 compared to Lite-HRNet30, 1.2 compared to HRFormer-T, and 6.5 compared to Token-Pose-T (pure transformer sturcture). Compared with regression-based methods, our method was the best, with the exception of ResNet101.

What’s more, it is worth noting that the AP50 of our method is even the best among all methods mentioned above. It is not difficult to find that our method occupies the absolute predominance in the speed test, due to the replacement of the self-attention module with the efficient and computationally friendly shift operator. What’s more, our model uses less video memory in Figure 6, which is very important on resource-constrained edge devices.

Therefore, the experiment shows that our method achieves outstanding performance on the COCO dataset.

#### 4.3.2. Result in MPII Dataset

In Table 2, ShiftPose-T archived 75.5 PCKh, ShiftPose-M archived 83.7 PCKh, ShiftPose-L archived 86.4 PCKh. Interestingly, we found that, on the COCO dataset, our method performed better than simple baseline ResNet50 and 101; however, it was the opposite situation on the MPII dataset. We attribute this interesting finding to the overfitting of ResNet50/101 on the MPII dataset, because the COCO dataset has more images than MPII. Additionally, this indirectly proves that our method has good robustness and generalization. And the visual results for MPII dataset are shown in Figure 7.

What’s more, our model provides the fastest speed among the mentioned models. Speed and accuracy have become our most obvious advantages.

In Figure 8, the ShiftPose-L cost the least time for convergence, and the ShiftPose-M is very close to the ShiftPose-L in the early stage. The ShiftPose-T takes the most time for convergence, and the difference between ShiftPose-T and ShiftPose-M is in the numbers of layer. ShiftPose-T obtains 2×5 = 10 layers and ShiftPose-M obtains 4×5 = 20 layers. Therefore, we can draw a conclusion: increasing the layers of each stage gains more profits than increasing the dim of input channels. The ShiftPose-L had 10.2M parameters, with 86.4 PCKh, and ShiftPose-M had only 4.16M parameters, with 83.7 PCKh by comparison.

#### 4.3.3. Result on Human3.6M Dataset

The results of the Human3.6M dataset are shown in Table 3. ShiftPose-L contains half of the parameters of RestNet50 and costs one-third GFLOPs. And the visual results are shown in Figure 9.

In Table 3 and Table 4, we can find two strange phenomena:Both our model and ResNet50 obtained a lower error in X and Y than Z.Our model performed better than ResNet50 in 2D pose estimation, but the opposite was true in 3D pose estimation.

From phenomena 1, we can easily find that the depth estimation was harder than 2D human pose estimation. As for phenomena 2, noticing that our model performed better than ResNet50 in 2D human pose estimation, while having worse error-x and error-y in 3D human pose estimation, we conjecture that the lightweight model (ours) tends to centralize computing resources to exploit effective representation for 2D human poses. After introducing a 3D pose estimation task, a portion of the computing resources had to be used for depth estimation, thus resulting in decreased accuracy of the 2D human pose estimation. We tested our hypothesis by simply changing the patch size of our neural network from 4 to 2, changing the dimension from 64 to 128, and keeping the structure the same.

The ShiftPose with more parameters worked as expected, and we can draw three conclusions from the experiments:The computer resource of ShiftPose-L (dim = 64) has been fully used for 2D pose estimation.Limited by the number of parameters, the lightweight model’s capacity for 2D pose estimation began to weaken, while exploiting the depth representation.It is better for the lightweight model to predict 3D poses than 2D poses because, generally, the model using 2D poses to predict 3D poses, such as Pose Lift [37], are also lightweight.

In order to keep the model lightweight and single-stage, we should pay more attention to optimizing the structure or designing a new structure, so that the model can learn a better representation, without any extra parameters. It will be investigated in more detail in future work.

### 4.4. Ablation Study

#### 4.4.1. Plain and Bridge–Branch Structure

To evaluate the effectiveness of the bridge–branch structure, we directly shrunk the branch in ShiftPose-L and changed it into a single branch architecture (named Plain architecture). In Table 5, the bridge–branch architecture obtained an increase of 4.8 PCKH, which was more than plain architecture.

#### 4.4.2. Replacement of Shift Block

In Table 6, after removing the shift operator, the PCKh dropped rapidly; however, when we replace the shift operator with the W-MSA module, the model stops learning, and the PCKh remains slightly higher or lower than 17.5. This is quite out of our expectation, compared with the model with the attention module, so we added the result of the pure transformer: TokenPose-S in Table 7. We guess that the reason for this is maybe the W-MSA in the Swin transformer is unsuited to the human pose estimation task.

#### 4.4.3. Improved RLE

Limited by the hardware support, we only test three values of σ = 1, 1.5, 2.5, and the result is indeed influenced by σ, when σ=2.5, the model gets the best performance. The detailed results are shown in Table 8. In addition, the RLE is not robust to wrong annotations and easily gets crashed without a good initialization and batch normalization.

.

## 5. Discussion

### 5.1. 3D Pose Estimation

Depth estimation is dependent on the features extracted from the backbone and interaction between features. So, large models, such ResNet and HRNet, with large parameters and feature fusion operations, do not worry about this question; however, it is the opposite situation in the lightweight model. With limited computation resources, the lightweight model would like to concentrate resources to explore high-level semantic information, which brings high benefits, instead of wasted computation in multi-branch structure. Furthermore, our experiments confirmed this point of view and drew an expanded conclusion on 3D human pose estimation.

The experiments showed that our model with bridge–branch structure can handle the 2D human pose estimation task well, but it is not good at 3D human pose estimation. After exploring the results of 3D human pose estimation in depth, we find that the model has already performed well on X- and Y-coordinates; therefore, we can infer that the ability of representation of the model shifts away from 3D human pose estimation towards 2D human pose estimation, and our model needs a more efficient regression head to generate more accurate Z-coordinates.

The weakness of 3D human pose estimation inspires us to improve the ability of extracting a good depth representation from a single image. First, we can increase the number of parameters and introduce auxiliary supervision, such as ordinal ranking, to alleviate the difficulty of learning depth information; second, the current regression head is quite simple, without any explicit or implicit feature interaction between 2D representation and depth representation. Therefore, redesigning the regression head may help a lot.

### 5.2. Stable Training on RLE

In our experiments, we found that, when we simply applied the ShiftPose with RLE to 3D human pose training, the loss became very large and easily crashed. To prevent this strange error, we added a batch normalization on the final layer, before the regression head; after that, we never met the same solution. Additionally, RLE loss is not robust for wrong annotations; so, if your method with RLE crashed in the experiments, this may help you to solve the problem.

Meanwhile, for stable training, we attempted to make three dimensions (x,y,z) share the same sigma in the training process, which, indeed, helps training.

### 5.3. Optimize ShiftPose

Our model’s configuration is maybe not the best, but the validity of our model was evaluated by the experiments, and future work can focus on the design of the hyper parameter with neural architecture search technology [54] and deploy the ShiftPose to a real-time pose estimation system, such as AlphaPose [55].

As for the regression head, we do not design it on purpose in order to prove the strength of our backbone, and in the experiments, we observed that the ShiftPose cannot recognize the ankle, keen, and wrist, to enhance the ability of learning these parts, you can design a new efficient regression head to get a better performance.

### 5.4. Bottom-Up Method

Although the top-down method has higher accuracy in human pose estimation, but in multi-person situation, the human detector costs lots of time in detecting humans before human pose estimation, while the bottom-up method removes the detector and predicts the coordinates directly, which is much faster than top-down method, theoretically speaking.

Future works can focus on the improvement of the speed and apply the ShiftPose as a strong backbone to multi-person pose estimation tasks, naturally, we recommend the bottom-up method, which can archive faster speed and have great potential.

## 6. Conclusions

Among the regression-based methods, our model, named ShiftPose, obtained excellent performance, with much higher fps, compared with some of the transformer-based methods. What’s more, our method was even competitive, compared with the heatmap-based methods, which proves that the strength of the transformer architecture and self-attention module has little contribution to the human pose estimation task.

We provide three kinds of ShiftPose: ShiftPose-T, ShiftPose-M, ShiftPose-L, and ShiftPose-L obtained 86.4 PCKH, 29.1 PCKH@0.1. On the MPII dataset, 72.1 mAP, 255 fps on the GPU, with 10.2M parameters, and 1.6 GFLOPs on the COCO dataset.

In the experiments, compared with results on the COCO and MPII datasets, our model was more robust and had better generalization than ResNet50/101. For the model itself, the bridge–branch structure performed better than the plain structure on all the datasets, and it gained more profit by increasing the number of layers in each stage than increasing the channel dimension of each stage, compared with the simplest configuration (ShiftPose-M). The former (ShiftPose-M) only increased the 1.3M parameters; however, the latter (ShiftPose-L) increased the 7.3M parameters.

In addition, interestingly, we find that lightweight models tend to learn x- and y-coordinates as priority in the 3D HPE training process. As a result, in order to improve performance on 3D HPE, the lightweight model need to increase the number of parameters, introduce extra auxiliary supervision, or redesign the regression head.

Finally, we discussed some interesting phenomenon during training process. We found that the RLE loss crashed easily without any normalization and proper initialization, and the accuracy of the model would stay at a small value. After we added a normalization layer, before the regression head, the training process became stable. What’s more, if we make the X-, Y-, and Z-coordinate share a same σ, the training process will also become stable.

In the future, we are going to apply our model to multi-person pose estimation and deploy it to resource-constrained edge devices, such as Raspberry Pi and mobile phones. To improve the speed and performance, we will use neural architecture search technology to optimize the structure and hyper parameters of our model.

## Figures and Tables

**Figure 1 sensors-22-07264-f001:**
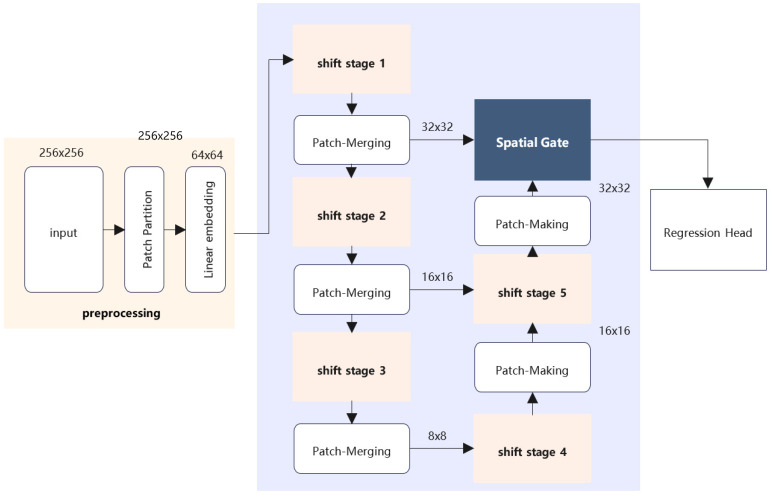
The architecture for ShiftPose.

**Figure 2 sensors-22-07264-f002:**
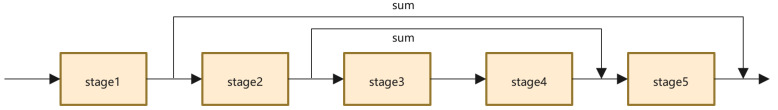
The bridge–branch architecture.

**Figure 3 sensors-22-07264-f003:**
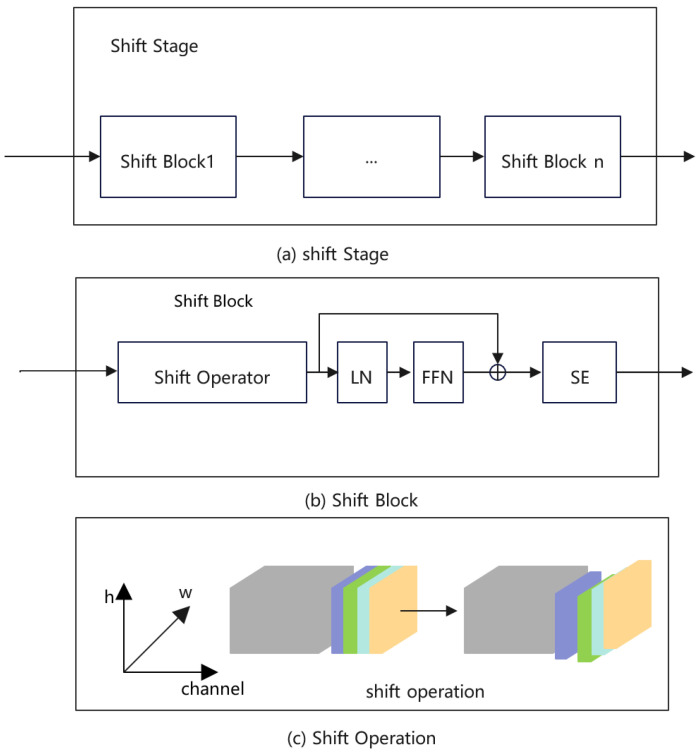
(**a**) The structure of shift stage; (**b**) the structure of shift block; (**c**) the implement of shift operation.

**Figure 4 sensors-22-07264-f004:**
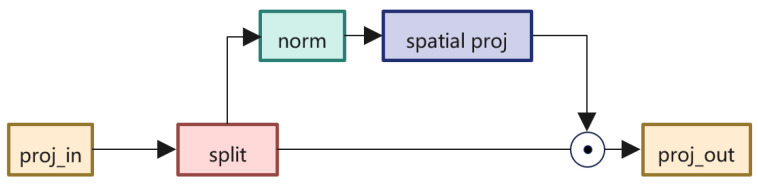
The spatial gate structure in our model. Both *proj_in* and *proj_out* are linear layer.

**Figure 5 sensors-22-07264-f005:**
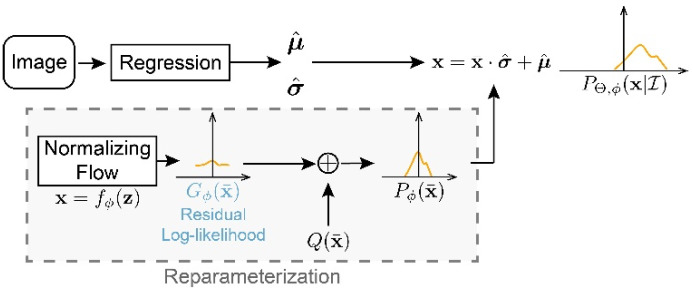
The process of residual log-likelihood estimation.

**Figure 6 sensors-22-07264-f006:**
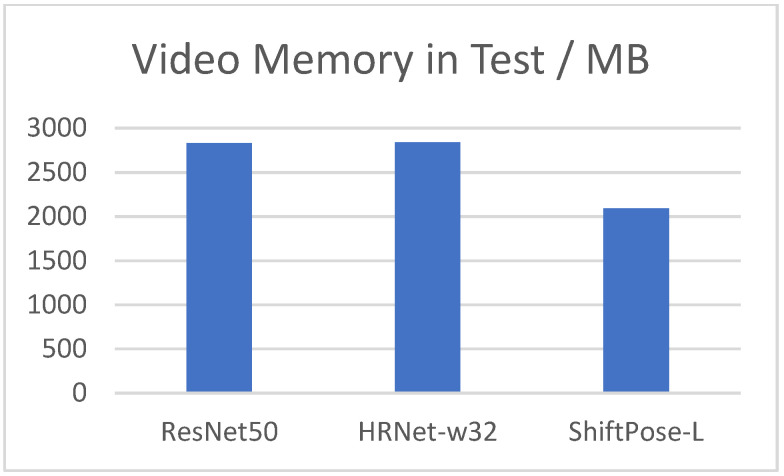
The video memory per GPU in testing, with batch size = 32.

**Figure 7 sensors-22-07264-f007:**
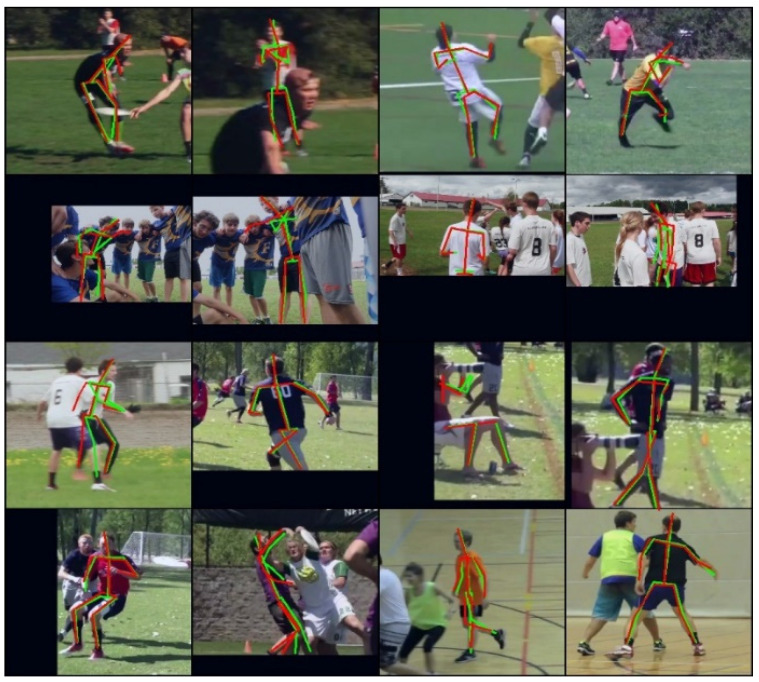
The predicted results on MPII dataset with ShiftPose-L, ground truth (red) and prediction (green).

**Figure 8 sensors-22-07264-f008:**
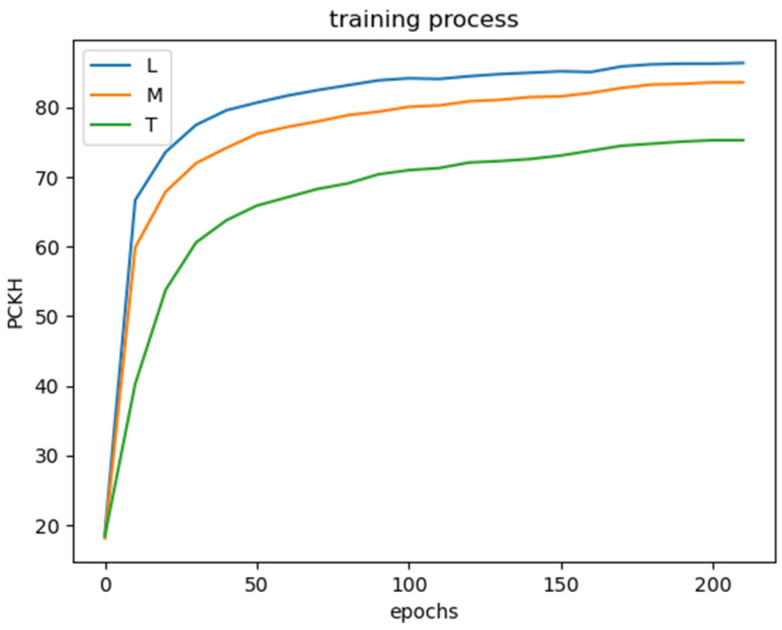
The training process.

**Figure 9 sensors-22-07264-f009:**
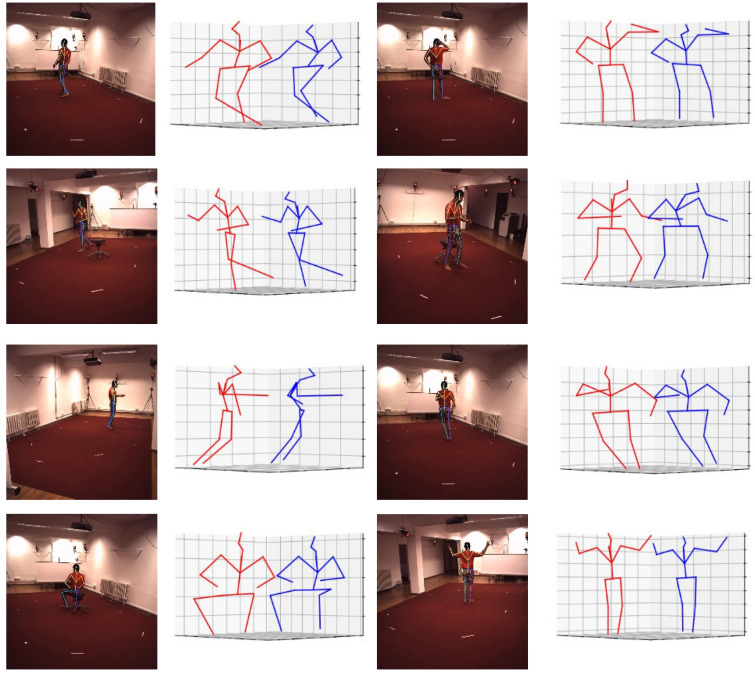
The predict results on Human3.6M dataset (red: ground truth; blue: predict).

**Table 1 sensors-22-07264-t001:** The results in COCO dataset.

Model Name	Input Size	Params	GFLOPs ^5^	AP	AP50	Speed GPU (fps) ^4^
**Heatmap-based**						
Simba-Res50 [5] ^1^	256 × 192	34M	8.9	70.4	88.6	48 ^2^
Simba-Res101 [5]	256 × 192	52.6M	9.3	71.4	89.3	-
HRNet-w32 [19]	256 × 192	28.5M	7.1	73.4	89.5	28 ^2^
Lite-HRNet30 [46]	256 × 192	1.8M	0.31	70.4	88.0	12 ^2^
HRFormer-T [32]	256 × 192	2.5M	1.3	70.9	89.0	-
HRFormer-S [32]	256 × 192	7.8M	2.8	74.0	90.2	-
Token-Pose-T [33]	256 × 192	5.8M	1.3	65.6	86.4	42 ^3^
OpenPose [47]	256 × 256			61.8	84.9	
Regression-based						
DeepPose [21]	256 × 256	23.6M	5.4	52.6	81.6	135 ^3^
Res50+RLE [18]	256 × 256	23.6M	5.4	71.3	88.9	135 ^3^
Res50+SimDR [53]	256 × 192	36.8M	9.0	71.4	-	120 ^3^
Res101+SimDR [53]	256 × 192	55.7M	12.4	72.3	-	-
ShifPose-L(ours)	256 × 192	10.2M	1.6	72.1	91.5	255 ^3^

^1^ Simple baseline ResNet; ^2^ tested on NVIDIA GTX 1660 SUPER and Intel(R) Core(TM) i7-8700 CPU @ 3.20 GHz; ^3^ tested on NVIDIA RTX A4000 and Intel(R) Xeon(R) CPU E5-2686 v4 @ 2.30GHz; ^4^ our method of computing fps did not include the data preprocessing. ^5^ Floating point operations.

**Table 2 sensors-22-07264-t002:** The results on MPII dataset.

Model Name	Input Size	Params	GFLOPs	PCKh	PCKh@0.1	Speed GPU (fps)
DeepPose [21]	256 × 256	23.58M	4.04	82.5	17.4	135 ^1^
Simba-Res50 [5]	256 × 256	34M	8.9	88.5	33.9	48 ^1^
Simba-Res101 [5]	256 × 256	56.7	10.4	89.1	34.0	-
HRNet-w32 [19]	256 × 256	28.5M	17.3	92.3	-	-
Lite-HRNet30 [46]	256 × 256	1.8M	0.43	87.0	-	-
TokenPose-L/D6 [33]	256 × 256	21.4M	-	90.2	-	-
OpenPose [47]	256 × 256	-	-	75.6	-	200
ShiftPose-T	256 × 256	2.89M	0.69	75.5	17.0	945 ^2^
ShiftPose-M	256 × 256	4.16M	1.00	83.7	25.0	440 ^2^
ShiftPose-L	256 × 256	10.20M	1.63	86.4	29.1	308 ^2^

^1^ Test with batch size 32. ^2^ Test with batch size 128. We also performed experiments on MPII dataset, and we used an extremely simple structure in ShiftPose-T, with mlp_ratio = 1 in FFN module and input channel = 32, which also obtained a good performance.

**Table 3 sensors-22-07264-t003:** The results in Human3.6M dataset.

Model Name	Input Size	Params	GFLOPs	MPJPE	PA-MPJPE
ResNet50+RLE [18]	256 × 256	23.8M	5.40	48.6	38.5
ShiftPose-L+RLE	256 × 256	10.2M	1.64	73.4	53.3

**Table 4 sensors-22-07264-t004:** The detail results on Human3.6M dataset.

Model Name	PA-MPJPE	MPJPE	Error-X	Error-Y	Error-Z
ResNet50 [18]	38.5	48.6	16.0	16.0	37.1
ShiftPose-L (dim = 64, patch size = 2)	42.3	52.1	17.0	17.3	42.0
ShiftPose-L (dim = 64, patch size = 4)	53.3	73.4	21.5	24.4	57.7
ShiftPose-L (dim = 128, patch size = 4)	44.7	59.1	18.0	19.9	45.0

**Table 5 sensors-22-07264-t005:** Comparison of the plain and bridge–branch structures.

Model Name	GFLOPs	PCKh	PCKh@0.1
Plain	1.00	81.6	21.5
Bridge	1.63	86.4	29.1

**Table 6 sensors-22-07264-t006:** The results for different modules (none, self-attention, and shift) of MPII dataset.

Module	PCKh	PCKh@0.1
Without shift	25.4	1.4
With W-MSA	17.5	0.8
With shift	86.4	29.1

**Table 7 sensors-22-07264-t007:** The results of COCO dataset.

Model Name	AP	Speed GPU (fps)
TokenPose-S + SimDR [53]	73.6	80
Ours	72.1	308

**Table 8 sensors-22-07264-t008:** The results for different σ.

σ	MPJPE	PA-MPJPE
1	71.0	55.3
1.5	70.2	53.9
2.5	66.3	51.4

## Data Availability

The data presented in this study are available on request from the corresponding author.

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
