# Peer review of "Shift Pose: A Lightweight Transformer-like Neural Network for Human Pose Estimation"

_sensors, 2022, doi:10.3390/s22197264_

Round 1
Reviewer 1 Report
This paper proposes a Transformer-based model without self-attention module named Shift Pose for human pose estimation. The proposed network replaces the self-attention module with a non-parameter operation called shift operator to reduce the number of parameters and computational costs. Also, the network adapts the bridge branch connection instead of the fully-branch connection as a multi-resolution integration scheme.
Some detailed comments are listed below:
1. It is not quite clear whether the main contribution of this paper is for 2D or 3D human pose estimation. These two topics focus on different key points and to my knowledge, there is no universal method for both topics. Especially, according to the results of 3D estimation in the paper, the accuracy is not competitive enough compared to recent 3D estimation works.
2. The Transformer is a well-known model for the self-attention mechanism. So why is the proposed network which is without the SA module can still be considered as a Transformer-based network? Besides, if the replacement of the SA module has advantages, they should be presented in the ablation study or other comparative experiments.
3. The citation of the baseline methods in the experiment section should be added.
4. The ground truth of the Human3.6M results should be presented for comparison.
5. The manuscript needs to be checked carefully. There are many typos, grammar mistakes, and editing mistakes, such as:
line 22 Simple->simple
line 279 What's more, It is ... -> it
line 168 The Patch->patch
There are two section 3.2 in the manuscript, line 147 and line 157.
Different fonts: title of the paper, line 133 section 3.1, line 361 section 5.1, line 383 section 5.2
Some figures need to be modified, such as figure 1: pre-process, figure 3 (b): the arrow of the skip connection, (c) w is hidden, chan- ->channel, part of the subtitle is hidden.
There are many mistakes of the third person singular, especially in the related work section such as [21] firstly use..., [12] propose..., [33] first apply..., [32] replace.., [31] explore -> explores, [31] propose and so on.
And it looks strange to use the sequence number of literatures as the beginning of paragraphs or sentences.
Author Response
Dear reviewer,
Thank you very much for your comments and professional advice. These opinions help to improve academic rigor of our article. Based on your suggestion and request, we have made corrected modifications on the revised manuscript. as Re:reviewer1.pdf

Reviewer 2 Report
This paper proposed a Transform-based model without a self-attention module for human pose estimation. The article may be of interest to the reader but requires some corrections.
1) All abbreviations should be explained before the first use (e.g. the paper does not explain what PCKH means).
2) Please correct the errors in the drawings.
3) The paper requires thorough proofreading.
4) The paper should explain how the data sets were divided into training and test data.
5) "our method of computing fps doesn’t include the data preprocessing. " - Why does it not include preprocessing? If it did, would the proposed method be worse than others?
6) Why is the skeleton not shown in the MPII set (as for Human3.6M)? For the joints themselves, it is hard to tell if the result is correct.
7) Why do the results for the MPII and Human3.6M sets contain so few models? The method set should be as for the COCO dataset.
8) The literature review should also mention pose estimation methods that do not use deep learning methods.
Author Response
Dear reviewer,
Thank you very much for your comments and professional advice. These opinions help to improve academic rigor of our article. Based on your suggestion and request, we have made corrected modifications on the revised manuscript. it is show as Re:reviewer2.pdf.

Round 2
Reviewer 2 Report
I still have some comments:
1) there are still mistakes in the paper, e.g. there should be a space before the quotation brackets
2) Your answer to comment 5 ("We apologize for our vague ...") - you should explain this in the article.
3) "However, we don’t have enough time to finish this work, so we leave this work for future researchers." - never write that you did not do something because you did not have time. Write that it will be investigated in more detail in future work.
Author Response
Dear reviewer,
We really appreciate you for your carefulness and conscientiousness. Your suggestions are really valuable and helpful for revising and improving our paper. According to your suggestions, we have made the following revisions on this manuscript:
1.there are still mistakes in the paper, e.g., there should be a space before the quotation brackets.
Response: As suggested, we add a space before the quotation brackets, and we also correct some spelling and grammar mistakes. The revisions made to the manuscript are marked up using the “Track Changes” function in MS Word.
2.Your answer to comment 5 ("We apologize for our vague ...") - you should explain this in the article.
Response: We thank the reviewer for the suggestion and add the explanation to our paper in line 284:
Some methods have different or special data preprocessing methods, although the difference between different preprocessing method is not great, for the sake of fairness, we only compare the time of inference with other methods.
3."However, we don’t have enough time to finish this work, so we leave this work for future researchers." - never write that you did not do something because you did not have time. Write that it will be investigated in more detail in future work.
Response: We thank the reviewer for pointing this out and have updated the relevant statement.
Original: “If you want to keep the model lightweight and single-stage, we recommend you to optimize the structure of model in order to learn a better representation without adding extra parameters and design a new structure. However, we don’t have enough time to finish this work, so we leave this work for future researchers.”
Edited: “In order to keep the model lightweight and single-stage, we should pay more attention to optimize the structure or design a new structure so that the model can learn a better representation without any extra parameters. And it will be investigated in more detail in future work.”
Finally, we would like to thank the referee again for taking the time to review our manuscript.
